# What are the consequences of caring for older people and what interventions are effective for supporting unpaid carers? A rapid review of systematic reviews

Gemma F Spiers [1], Jennifer Liddle [1,2], Tafadzwa Patience Kunonga [1], Ishbel Orla Whitehead [1], Fiona Beyer,[1] Daniel Stow [1], Claire Welsh [1], Sheena E Ramsay,[1] Dawn Craig,[1,2] Barbara Hanratty [1,2]

¹Population Health Sciences Institute, Newcastle University, Newcastle upon Tyne, UK
²NIHR Applied Research Collaboration North East & North Cumbria, Newcastle upon Tyne, UK

**Correspondence to**
Dr Gemma F Spiers;
gemma-frances.spiers@newcastle.ac.uk

## ABSTRACT

**Objectives** To identify and map evidence about the consequences of unpaid caring for all carers of older people, and effective interventions to support this carer population.

**Design** A rapid review of systematic reviews, focused on the consequences for carers of unpaid caring for older people, and interventions to support this heterogeneous group of carers. Reviews of carers of all ages were eligible, with any outcome measures relating to carers' health, and social and financial well-being. Searches were conducted in MEDLINE, PsycInfo and Epistemonikos (January 2000 to January 2020). Records were screened, and included systematic reviews were quality appraised. Summary data were extracted and a narrative synthesis produced.

**Results** Twelve systematic reviews reporting evidence about the consequences of caring for carers (n=6) and assessing the effectiveness of carer interventions (n=6) were included. The review evidence typically focused on mental health outcomes, with little information identified about carers' physical, social and financial well-being. Clear estimates of the prevalence and severity of carer outcomes, and how these differ between carers and non-carers, were absent. A range of interventions were identified, but there was no strong evidence of effectiveness. In some studies, the choice of outcome measure may underestimate the full impact of an intervention.

**Conclusions** Current evidence fails to fully quantify the impacts that caring for older people has on carers' health and well-being. Information on social patterning of the consequences of caring is absent. Systematic measurement of a broad range of outcomes, with comparison to the general population, is needed to better understand the true consequences of caring. Classification of unpaid caring as a social determinant of health could be an effective lever to bring greater focus and support to this population. Further work is needed to develop and identify suitable interventions in order to support evidence-based policymaking and practice.

## BACKGROUND

Populations are ageing worldwide. Life expectancy is increasing in many high-income

### Strengths and limitations of this study

► A rapid review of published systematic reviews offers a useful and efficient approach to summarising evidence about caring for older people, without duplicating existing work.
► This approach allowed us to identify key gaps in this evidence base and make clear recommendations for future research.
► A limitation of this method is the exclusion of primary research studies that have not yet been subject to a systematic review.
► Most of the included systematic reviews were published within the last 3 years, signalling a contemporary evidence base.

countries, while the proportion of those aged 65 and over is projected to increase 16% by 2050.[1] However, longer lives will not necessarily be spent in good health, and population ageing will likely result in an increase in care needs. Indeed, current projections suggest that the number of dependent older people in the UK will increase 113% by 2051.[2] Current state provision of social care has failed to keep pace with current levels of demand.[3 4] For example, around 1.5 million older people are estimated to have an unmet need for care.[5] The consequences of this gap between care needs and care provision are also likely to be experienced by families and friends who provide unpaid care. Indeed, there are approximately 6.5 million people in the UK providing unpaid care to ill and disabled family members, friends or partners,[6] with an estimated economic contribution of up to £132 billion/year.[7] In this paper, we use the term 'carer' to refer to people providing unpaid care, although we recognise that not every carer identifies themselves as such.[8]

Carers make a critical contribution to health and social care.[9] Yet caring can have profound impacts on the lives of carers. The demands of caring bring emotional, health and financial challenges,[10 11] alongside changes to home and daily routines, work and career, family and social networks.[12–14] Carers are at greater risk of premature death and higher disease prevalence, while neglect of their own healthcare needs is common.[15] Such poor health is likely to be exacerbated by the social isolation, poor information and support, and financial stress that carers experience.[16] Supporting the mental and physical health and well-being of carers is essential. It is also critical that any form of support must account for the complexity of caring. For example, six key attributes of the caring experience (for those caring for an older person) have been identified as: getting on with the care recipient; assistance from organisations and the government; support from family and friends; activities outside caring; control; and fulfilment.[17] These attributes reflect the need to consider a broad range of outcomes when designing, implementing and evaluating carer support interventions.

Recognition of the role and contribution of carers has grown since the 1960s.[18] In the UK, the National Institute for Health and Care Excellence guidelines for supporting carers were published in 2020, highlighting this population as a policy priority.[19] Evidence-informed practice will ensure that the most effective interventions are used to support carers, while targeting the most relevant outcomes. As the number of older people requiring care rises, it is important that evidence is relevant to the contribution and support needs of people caring for older populations. Recent umbrella reviews have focused on evidence on interventions for carers more broadly or carers of people with dementia.[20–23] While this is important, there is now a clear need to map the evidence about carers of older people, with a focus on both the consequences of caring *and* the most effective forms of carer support. This will enhance our understanding of how best to support carers of older people.

We undertook a review of systematic reviews to map current evidence about caring for an older person, addressing two key questions:

1. What are the consequences of caring for older people for the health, and social and financial well-being of carers, and what do we know about how these consequences vary by age, sex, socioeconomic status and geographical location?
2. Which interventions are effective (including consideration of costs) to promote health and well-being and access to services among these carers?

The aim of this work was to provide an overview of what is known, as well as identifying key gaps, to support evidence-informed practice for those caring for older people.

## METHODS

The approach to this evidence synthesis was a rapid review of published systematic reviews (from hereon referred to as 'rapid review'). Preliminary scoping of the literature identified multiple systematic reviews on both the consequences of caring and associated interventions to support those caring for older people. Thus, the review of reviews ('umbrella review') approach was most appropriate as an efficient approach to assessing the evidence, without duplicating existing research. Rapid review methodology was employed, which uses a streamlined approach to study selection and synthesis in order to produce a timely overview of evidence.[24 25]

The following methods are reported in accordance with the Preferred Reporting Items for Systematic Reviews and Meta-Analyses guidelines.

### Search strategy

A search strategy was developed, piloted and refined, combining terms for 'carers' with validated systematic review filters (see online supplemental materials).[26] Searches were carried out in MEDLINE, PsycInfo and Epistemonikos (https://www.epistemonikos.org/) in January 2020, and were limited to English language publications published after 2000.

### Review criteria

Table 1 outlines the review criteria. Eligible systematic reviews were those that: reported evidence about the consequences of caring, or the effectiveness of interventions for promoting carer health and well-being and access to services, where the care recipients were older people. Initial scoping indicated that many reviews *did not* specify an age threshold (eg, >65 years) for older care recipients. Therefore, no age criteria for care recipients were specified in this rapid review. Instead, reviews were included if the care recipient population was described as older people, or where the recipient population was likely to include older populations. In consultation with a study steering group, reviews that focused solely on carers of people with dementia were excluded, as evidence about dementia caregivers has been summarised in three recently published reviews.[20–22] This exclusion criterion avoided duplication of this recent work and ensured we addressed the research questions for carers of older populations more broadly. All carer populations were eligible with no limits on age and sex.

As the focus of the (second) review question was about interventions to promote *carers'* health, well-being and access to services, reviews of *joint carer and recipient* interventions were excluded. Eligible outcomes were any that related to carers' health, social and financial well-being and access to services.

Reviews published after 2000 in English were included, and reviews of studies with any study design were eligible. Reviews must have met at least three of the five Database of Abstracts of Reviews of Effects (DARE) criteria to be considered a systematic review, with criterion 3 being

**Table 1** Review criteria

| | Inclusion criteria | |
| --- | --- | --- |
| | Synthesis 1: consequences of caring | Synthesis 2: interventions for carers |
| Population | Carers (eg, unpaid, family, 'informal') of older adults. No age criteria for care recipients are specified, but must be, or likely to include, older populations (eg, people with dementia). Carers include people of all ages, male and female. Care recipients (ie, older adults) include male and female, with any medical diagnosis, impairment, disability or frailty, and no limits to ethnicity. Setting—care provided in the community, hospital, care home. | |
| Intervention | Not applicable. | Any carer intervention that is targeted only on the carer (ie, not a joint carer and care recipient intervention) and which aims to improve carers' health, well-being and/or access to services. |
| Comparator | No comparator, or non-carers. | Any or no comparator, including usual care. |
| Outcome | Health status, quality of life, well-being, incident ill health, admission to hospital, financial well-being, poverty, measured changes in material circumstances, social relationships including loneliness, isolation, social support, social networks. | |
| Study design | Systematic reviews (those that meet 3 of 5 DARE criteria). Publication dates 2000–2019. If more recently published systematic reviews include evidence that is also in earlier reviews, the most recent reviews will be prioritised to avoid duplication. English language publications. | |

DARE, Database of Abstracts of Reviews of Effects.

mandatory.[27] Four out of five DARE criteria are the usual threshold to be considered a systematic review. However, this was lowered to three criteria here to maximise the number of reviews captured and thus evidence scoped. Criterion 3 was mandatory, as quality assessment of included studies is a key component of identifying the most robust evidence.

### Study selection
Records were managed in Rayyan software (https://rayyan.qcri.org), an online platform to manage and assist screening for systematic reviews.[28] Titles and abstracts were screened for relevance: 20% by two researchers independently, and the remaining 80% by a single researcher, as per rapid review methods.[24] For the 20% of records screened jointly, disagreements were resolved by taking forward any records, for which decisions differed, to the next stage of screening. Records that were deemed relevant based on title and abstract were retrieved for full-text screening against the review criteria by one researcher.

### Data extraction
Separate data extraction forms were developed and piloted for each review question using an Excel spreadsheet. Review details and characteristics were extracted, including: author, year of publication, number of studies included in the review, carer and care recipient population, type of consequence reported, interventions evaluated, outcomes and type of synthesis. Review findings and conclusions were summarised, including any subgroup analyses where reported.

### Quality appraisal
A full risk of bias assessment is not typically part of a rapid review.[24 25] However, an *indication* of bias was deemed important for this synthesis in order to identify the most robust findings, and obtain an overview of the quality of the evidence base. Thus, an abbreviated risk of bias appraisal was completed for included reviews using an amended version of the Risk of Bias in Systematic Reviews tool.[29]

This appraisal used five items relating to review methodology that indicate likely risk of bias (online supplemental table S1). Each item was scored 1 for 'yes' and the total score was summed across the five items. A score of 5 indicates a low risk of bias; 4 indicates a moderate risk of bias; 3 or less indicates a high risk of bias.

### Data synthesis
A narrative synthesis was used to address each question, with reviews grouped into those reporting caring consequences (question 1) and those reporting effectiveness of carer interventions (question 2). Summary tables were produced to describe review characteristics, the type of evidence identified and indicative risk of bias. Findings were synthesised: first by the judged risk of bias (ie, low, moderate and high risks of bias) to prioritise the most robust evidence; and then by the type of consequence (question 1) and type of intervention (question 2). For review question 2, all identified interventions from included reviews were mapped against the outcomes reported to identify key gaps in evidence.

### Patient and public involvement
A steering group representing diverse perspectives and organisations provided advice on the content of the

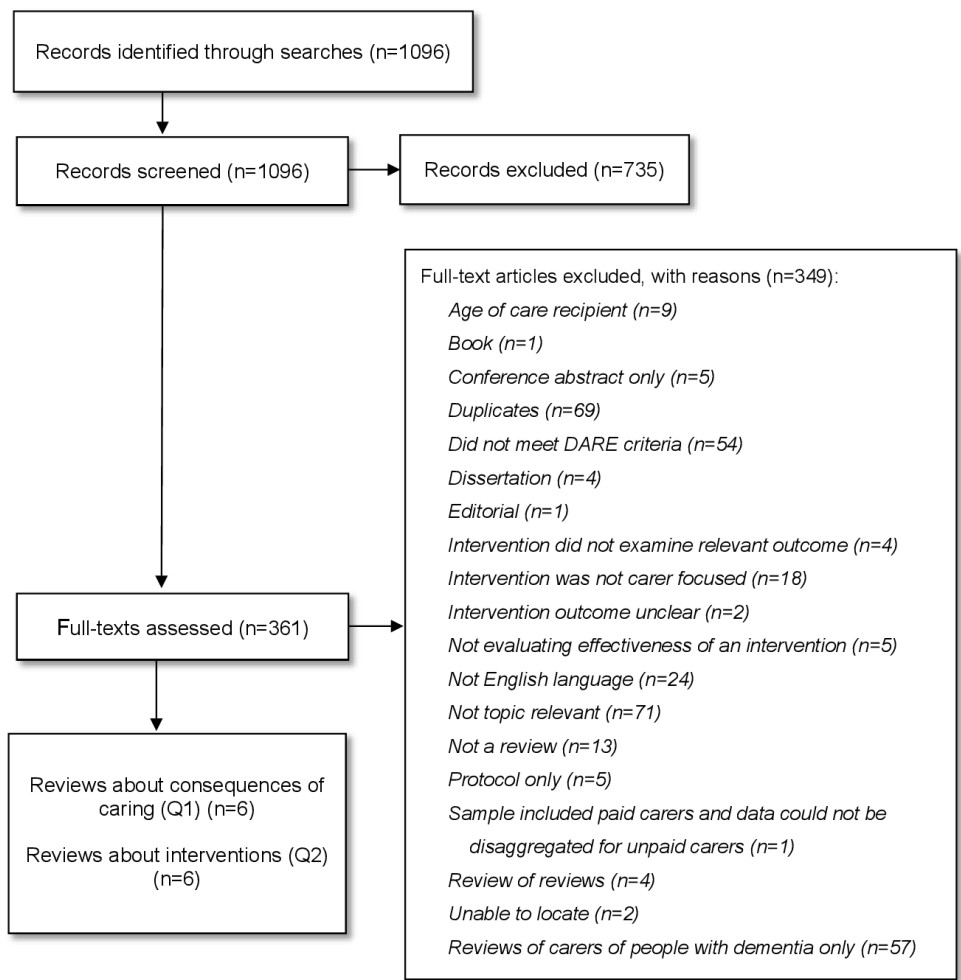

**Figure 1** Preferred Reporting Items for Systematic Reviews and Meta-Analyses (PRISMA) flow chart. DARE, Database of Abstracts of Reviews of Effects.

review and dissemination in regular meetings. In particular, we excluded carers of people with dementia from this study after discussion with them.

## FINDINGS

Twelve systematic reviews met the review criteria (figure 1). Six reviews reported evidence about the consequences of caring (table 2)[30–35] and six reported evidence about carer interventions (table 3).[36–41] In this section, we present the synthesis of evidence about the consequences of caring for older people (question 1) and the effectiveness of carer interventions (question 2). Evidence is reported according to judged risk of bias to prioritise the most robust findings.

## Consequences of caring

Of the six systematic reviews reporting evidence about the consequences of caring for older people: none were judged to have a low risk of bias; two a moderate risk of bias[32 35]; and four a high risk of bias[30 31 33 34] (online supplemental table S2).

### Moderate risk of bias

Two reviews reported that carers of older people experience 'burden'[i], depression and anxiety, but prevalence and severity were either not quantified, or estimates varied substantially.[32 35] For example, one review reported the prevalence of 'burden' among carers ranged from 1% to greater than 35%,[35] while the other reported estimates ranging from 37% to 100%.[32] The latter review presented limited evidence on which groups may be at greatest risk—this suggested that carers who are younger, male and with poor social and financial support may experience higher levels of 'burden'.[32] However, this evidence was not quantified in the review and so it is not possible to describe the difference in reported 'burden' between these groups.

### High risk of bias

Four reviews reported that carers of older people experience 'burden', anxiety and stress, but prevalence

---

[i]Due to insufficient detail reported in publications about the outcomes 'carer burden' and 'role strain', we are unable to provide a clear definition of these outcomes.

**Table 2** Overview of systematic reviews reporting evidence about the consequences of caring

| Study | Date | Studies in review (n) | Data range of included studies | Population (care recipient) | Type of consequence/ impact reported | Do studies within the review compare presence/ level of impact with non-carer samples? | Does the review identify evidence about impact for specific carer subgroups relating to age, sex, socioeconomic status and geographical location? | Synthesis | Indicative risk of bias |
|---|---|---|---|---|---|---|---|---|---|
| Amer Nordin et al[30] | 2019 | 19 | 1990–2017 | Older people | 'Caregiver burden', quality of life, perceived difficulty assisting recipient | No | No | Narrative | High |
| Bom et al[31] | 2019 | 15 | 2009–2017 | Older people | Mental health, physical health | Yes, controls matched for caregiver health | Sex, SES, location (European region) | Narrative | High |
| Ge and Mordiffi[32] | 2017 | 7 | 1999–2014 | Older people with cancer | 'Caregiver burden' prevalence and severity | No | Sex, SES, age | Narrative | Moderate |
| Jansen et al[33] | 2018 | 14 | 1994–2016 | Older cancer survivors | Prevalence and predictors of 'caregiver burden', depression, anxiety, self-esteem, distress, communication issues, stress and QoL | Yes, general population | Sex, SES | Narrative | High |
| Namasivayam-MacDonald and Shune[34] | 2018 | 4 | 2002–2017 | Older people | 'Caregiver burden' associated with feeding difficulties | No | No | Narrative | High |
| Ringer et al[35] | 2017 | 9 | 1997–2012 | Older people with frailty | 'Caregiver burden', negative reactions to caregiving | Unclear | No | Narrative | Moderate |

QoL, quality of life; SES, socioeconomic status.

**Table 3** Overview of systematic reviews reporting evidence about carer interventions

| Study | Date published | Studies in review (n) | Data range of included studies | Population (care recipient) | Interventions evaluated | Outcome | Synthesis | Indicative risk of bias |
|---|---|---|---|---|---|---|---|---|
| Domingues et al[36] | 2018 | 4 | 2013–2016 | Older people with mild cognitive impairment | Cognitive, multicomponent | 'Burden', quality of life, anxiety, mood | Narrative | High |
| Guay et al[37] | 2017 | 12 | 2000–2015 | Older people | Internet-based interventions, including education, self-help therapeutic and human-supported therapeutic | Depression, anxiety, stress, 'strain', 'burden', physical burden, self-perceived health, quality of life | Narrative | High |
| Lopez-Hartmann et al[38] | 2012 | 10 | 2002–2009 | Older people | Individual and group psychosocial support | Depression, 'burden', stress, 'role strain', coping, knowledge, social support, anxiety, economic burden | Narrative | High |
| Mason et al[39] | 2007 | 22 | 1985–2003 | Older people | Respite | Satisfaction, 'burden' | Both | Low |
| Murfield et al[40] | 2019 | 4 | 2013–2018 | Older people | Mindfulness+stress reduction, yoga and meditation | Self-compassion, 'other health' outcome | Narrative | Moderate |
| Shaw et al[41] | 2009 | 104 | 2005–2009 | Older people | Respite | 'Burden', depression, anxiety, morale, anger, hostility, caregiving relationship | Both | Low |

and severity were either not quantified or highly variable.[30 31 33 34] Evidence about the severity of depression among carers was also variable.[30 31 33] For example, one review reported evidence of mild, moderate *and* severe depression, although it was not clear how 'mild', 'moderate' and 'severe' were defined.[30] Another review reported that the impact on depression varied, but with no supporting detail.[33] One review indicated that levels of carer stress, anxiety and distress were higher than those of the general population, although by how much was not reported.[33]

There was limited and mixed evidence about the consequences of caring for physical health in one review, with a positive impact on self-rated health but also evidence of increased pain and medication usage.[31] Evidence about which groups may be at greatest risk was reported in two reviews.[31 33] One suggested that the impact of caring on health was greater for females and married people.[31] Yet another reported mixed evidence about whether 'burden' was greater for male or female carers.[33]

### Carer interventions

Of the six reviews reporting evidence about interventions to support those caring for older people, two were judged to have a low risk of bias[39 41]; one a moderate risk of bias[40]; and three a high risk of bias[36–38] (online supplemental table S3).

### Low risk of bias

Two reviews reported evidence on respite interventions for carers of older people.[39 41] Carers valued respite, with high levels of reported satisfaction. Yet there was no strong evidence to suggest respite improved carers' mental health or 'carer burden'. There were gaps in the evidence about the impact of respite on many other types of outcomes (eg, physical health).

### Moderate risk of bias

Limited evidence was presented in one review for the ability of a mindfulness stress reduction intervention to improve carer depression and anxiety. However, this intervention had no impact on stress, health service use, quality of life or self-compassion. In the same review, a combined yoga and meditation intervention appeared to improve carer self-compassion and quality of life.[40]

### High risk of bias

Reviews judged to have a high risk of bias reported evidence about cognitive, psychosocial and education-based interventions.

A review judged to be at high risk of bias reported limited and inconsistent evidence that a cognitive-based intervention was beneficial to carers.[36] The components of this cognitive intervention included 'calendar training' and 'note taking', although the overall objective of this intervention was not clear.

Evidence for the impact of self-help and supported therapy on carer depression, anxiety and 'burden' was inconsistent.[37] Similarly, there was inconsistent evidence

about the effect of individual psychosocial support interventions on 'carer burden' and depression.[38] Such individual psychosocial support interventions may improve carer stress and economic burden but this evidence was only identified in a single study in one review. There was no evidence of an effect on the outcome 'role strain'. Group psychosocial support interventions may improve carer stress, coping and knowledge but evidence was mixed for the outcomes of depression, 'carer burden' and 'role strain'.[38]

Finally, one review reported evidence about web-based educational interventions. Components of these educational interventions reported in the review included nurse and peer-led support to answer questions by email, provision of information on websites and training in relaxation and exercise skills. There was no consistent evidence that these interventions were more beneficial than usual care.[37]

### Key gaps in evidence about carer interventions

Online supplemental table S4 maps intervention types against the outcomes reported in the reviews. Evidence for respite and psychosocial interventions was most wide ranging.[37–41] By contrast, there were notable gaps in evidence for cognitive, educational and multicomponent interventions. Outcomes reported less often were economic burden, relationships and physical health.

## DISCUSSION

Older people's need for unpaid care is predicted to rise with population ageing.[2] Supporting the health and wellbeing of people providing unpaid care for older people, and mitigating the social and financial impacts of caring, should be a public health priority. Our rapid review of systematic reviews mapped current evidence to provide an overview of what is known, as well as identifying key gaps, about the health, social and financial impact of unpaid caring for older people and how this group of carers can be best supported.

The challenges and demands of unpaid caring for older people are well documented.[10 11 13 14] However, the current evidence fails to fully quantify the consequences of caring for older people on carers' health and wellbeing. Clear measurement of outcomes, with comparison to the general population, is needed to allow the full extent of this impact to be understood. Furthermore, carers are not a homogenous population, and unpaid caring takes place in diverse circumstances. A greater focus on which groups of carers are most vulnerable to the impact of caring, and under which conditions, would further enhance our understanding of these issues.

The evidence reviewed here highlights gaps in our understanding and suggests that a wider consideration of relevant outcomes is needed. Research into the impact of caring and effectiveness of carer interventions must consider outcomes beyond mental health. In particular, physical health was largely neglected in these reviews

about caring for older populations. A bias towards mental and psychological, rather than physical, health outcomes has also been noted by others in evidence about the consequences of caring.[42–44] Other reviews about the impact of caring at all ages report that carers are at increased risk of musculoskeletal conditions, cardiovascular disease, generalised cognitive deterioration and poor sleep.[42 45–47] Yet there is a clear gap in these outcomes in systematic reviews that focus on caring for older people. Similarly, we know that unpaid caring can lead to a loss of employment or a reduction in working hours.[48–51] Some carers face the demands of managing caring responsibilities alongside employment.[52] There are likely to be related impacts on financial and social well-being, but these outcomes were also absent in this evidence. Future research must address the full breadth of potential consequences of being a carer, with consideration of these important but overlooked outcomes.

Another limitation of the current evidence base is the oft-used outcome 'carer burden'. This is an ambiguous and contentious term,[53] and input from our stakeholder group confirmed the unhelpful nature of this concept. We recommend that future evaluation and reviews avoid this outcome and focus on more meaningful and specific measures of impact.

A final limitation of the evidence relates to whether chosen outcomes are adequate to capture the likely impact of interventions. Poor choice of outcomes may underestimate or overlook the ways in which an

intervention can effectively support and benefit carers of older people. This was most apparent in the reviews of respite interventions. Despite respite being valued by carers, there was no evidence that it improved carers' mental health or 'carer burden'. Yet respite alone cannot be expected to improve mental health, with other therapeutic support necessary to address these sorts of needs. There may also be outcomes where appropriate respite, even in the short term, may show beneficial effects (eg, stabilising carers' physical illness or injury). Evaluations could offer a more meaningful understanding of how an intervention can support carers by giving greater consideration to the outcomes measured and the potential pathways to impact.

Overall, our review of reviews indicates that a more robust and comprehensive evidence base is needed to inform policy and practice for supporting carers of older people. One mechanism to achieve this is by conceptualising unpaid caring as a social determinant of health. Social determinants are factors relating to life conditions and circumstances, such as age, income and working conditions, that impact on health. To locate unpaid caring as a social determinant recognises that this responsibility is likely to place carers at greater risk of poor health. Conceptualising unpaid caring as a social determinant of health is possible with an adaption of Dahlgren and Whitehead's model (figure 2).[54] Working and living conditions are acknowledged as social determinants of health. Highlighting the similar role of unpaid caring

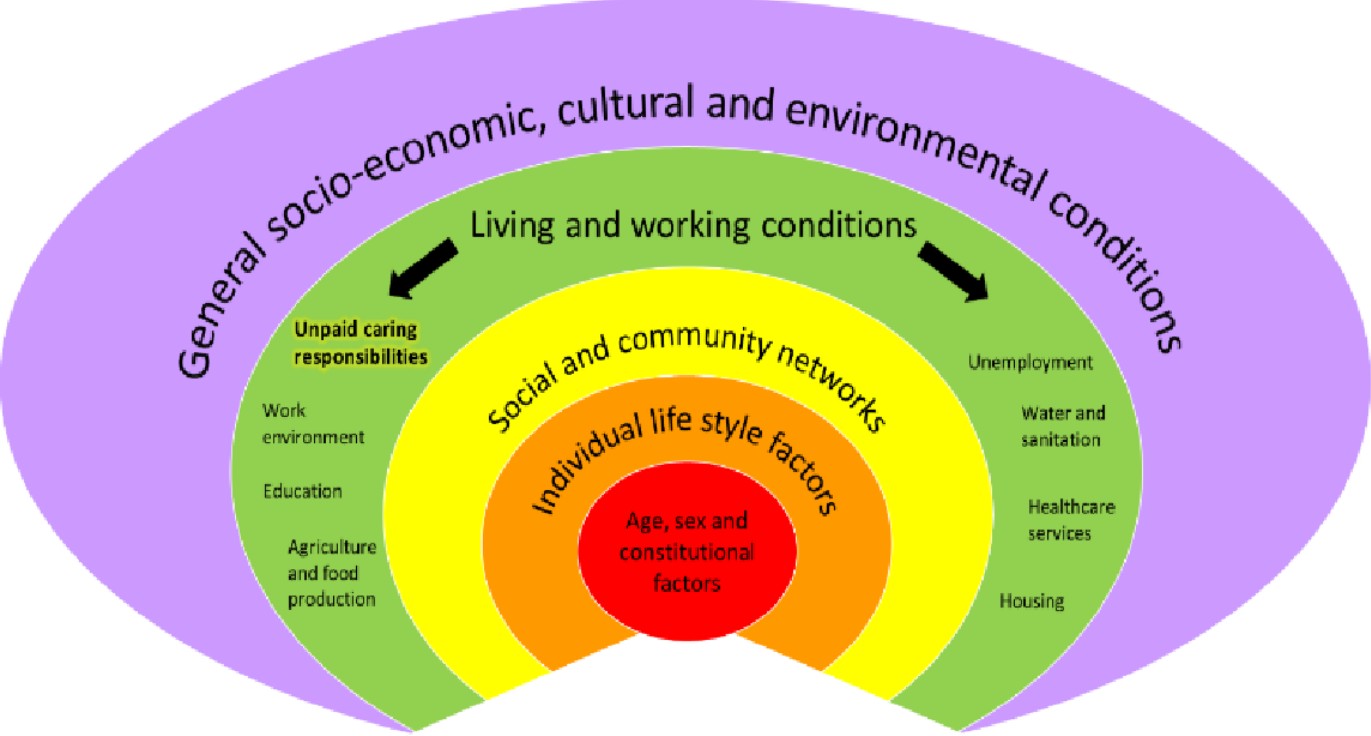

**Figure 2** Unpaid caring as a social determinant of health: an adaptation of Dahlgren and Whitehead's[54] model.

would be helpful here; caring responsibilities and their consequences shape health and are socially patterned. In an area of growing unmet need, framing unpaid caring as a social determinant of health may focus the attention of policymakers and drive the development of evidence-based interventions.

Moving forward, we propose the following future research priorities:

► A high-quality comprehensive systematic review about the impact of caring for older people on outcomes overlooked in current systematic reviews: physical health, and social and financial well-being. A systematic review of these outcomes should draw on evidence from non-peer-reviewed sources (eg, national statistical reports, third sector reports) and qualitative research to capture the full breadth of available evidence.

► Clear quantification of the prevalence and severity of mental health outcomes and other illnesses experienced by carers of older people, with comparisons to the general population to highlight the extent of any impact on health and well-being, and potential inequalities between carers and non-carers.

► Robust development and evaluation of promising interventions for carers of older people, with consideration of pathways to impact and a broader range of relevant outcomes.

The priorities outlined above have potential to address gaps in the current evidence base identified in our review, and to improve the overall quality and scope of evidence about caring for older people.

### Strengths and limitations

Our chosen method was a review of published systematic reviews to avoid duplicating existing work. This is a valuable approach that has highlighted key gaps in the evidence base about caring for older people. A limitation of this method is that primary studies, which have not yet been subjected to a systematic review, are inevitably excluded. However, nine of the 12 included systematic reviews were published within the last 3 years (2017–2019) and the remainder published between 2007 and 2012. The number of primary studies potentially excluded by using this method is therefore likely to be low.

### CONCLUSION

Current evidence about caring for older people lacks the necessary detail to fully quantify the impact of unpaid caring on carers, and to identify how this population of carers should be supported. Classification of unpaid caring as a social determinant of health could be an effective lever to bring greater focus to this population in an area of growing unmet need. Further research is needed to identify suitable interventions in order to support evidence-based policymaking and practice.

**Contributors** GFS and JL developed the protocol, undertook all stages of the review and cowrote the first draft of the paper. TPK developed the protocol, undertook all stages of the review and contributed to the final draft of the paper. OW developed the protocol, undertook record screening and data extraction for the review and contributed to the final draft of the paper. FB developed the protocol, undertook the study searches and contributed to the final draft of the paper. DS, CW and SER developed the protocol, undertook record screening for the review and contributed to the final draft of the paper. DC planned the study, developed the protocol, undertook record screening for the review and cowrote the first draft of the paper. BH planned the study, developed the protocol, undertook record screening and cowrote the first draft of the paper.

**Funding** This paper presents independent research funded by Public Health England (NU-002487 RES/0290/0170). JL and BH are funded by the National Institute for Health Research (NIHR) Applied Research Collaboration North East and North Cumbria (ARC NENC, NIHR200173).

**Disclaimer** The views expressed are those of the author(s) and not necessarily those of Public Health England or the NIHR.

**Competing interests** None declared.

**Patient consent for publication** Not required.

**Provenance and peer review** Not commissioned; externally peer reviewed.

**Data availability statement** Data sharing not applicable as no data sets generated and/or analysed for this study.

**ORCID iDs**
Gemma F Spiers http://orcid.org/0000-0003-2121-4529
Jennifer Liddle http://orcid.org/0000-0003-1059-1230
Tafadzwa Patience Kunonga http://orcid.org/0000-0002-6193-1365
Ishbel Orla Whitehead http://orcid.org/0000-0002-4171-8583
Daniel Stow http://orcid.org/0000-0002-9534-4521
Claire Welsh http://orcid.org/0000-0001-9477-0775
Barbara Hanratty http://orcid.org/0000-0002-3122-7190

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
