## [Reviewer comments · BMJ Open]

ARTICLE DETAILS

TITLE (PROVISIONAL)	What are the consequences of caring for older people and what interventions are effective for supporting unpaid carers? A rapid review of systematic reviews
AUTHORS	Spiers, Gemma; Liddle, Jennifer; Kunonga, Tafadzwa; Whitehead, Orla; Beyer, Fiona; Stow, Daniel; Welsh, Claire; Ramsay, Sheena; Craig, Dawn; Hanratty, Barbara

VERSION 1 – REVIEW

REVIEWER	van Marwijk, Harm University of Brighton, Division of Primary Care and Public Health
REVIEW RETURNED	11-Dec-2020

GENERAL COMMENTS	This paper aims to do a rapid review to identify and map evidence about the consequences of unpaid caring for all carers of older people, and effective interventions to support this carer population. I am not sure 'carers' would necessarily define themselves as such. It is one role among a number. It seems a little bit of a large aim, and slightly unspecified. The focus might be a little overly medical. What does it mean to care? A reflection on the wider dynamics around caring might help. It is probably not a unidimensional issue for most carers, there might be gender role issues etc.
--

REVIEWER	van Wyk, Neltjie University of Pretoria
REVIEW RETURNED	20-Jan-2021

GENERAL COMMENTS	Dear Authors, The
-------------------

REVIEWER	Giebel, Clarissa University of Liverpool, Institute of Psychology, Health and Society
REVIEW RETURNED	26-Jul-2021

GENERAL COMMENTS	This is an important topic, particularly in light of the pandemic, when more unpaid carers have to provide unpaid care for older adults, with a lack of social care services available. I only have a few comments, however, the last comment (all in chronological order) is major and needs specific attention: - The title should reflect that this is a review of SRs.
---

	 - Old age is classed as 65 and above (although that is incredibly outdated as we know...). however, please indicate why you have referred to age 85+ in the Introduction when referring to older adults. - There is a grammar mistake at the end of the intro "We *under* undertook...". - It seems strange that carers of people living with dementia were excluded from the study (specified under Public involvement). Could the authors please elaborate on this? Who made up the PPI group? Did it include carers, including those with dementia? - The authors very briefly mention Dahlgren and WhiteHEAD (spelling mistake again)'s model of health inequalities, yet fail to elaborate how this is relevant to the topic and the findings. The authors should elaborate on this further and thus strengthen this point. - If a more robust evidence base is needed, as outlined by the authors, how come a research priority is to conduct yet another SR? This would suggest more primary research is needed, to strengthen that evidence base. This seems particularly unnecessary as the authors state repeatedly that the included reviews were mostly conducted in the past few years, and thus conducting yet another SR would be a duplication of effort.
--	---

VERSION 1 – AUTHOR RESPONSE

Reviewer 1	Thank you for your constructive review of our paper. We have endeavoured to address your comments and we summarise our responses and revisions below. Edits are in red text.
-------------------	---

This paper aims to do a rapid review to identify and map evidence about the consequences of unpaid caring for all carers of older people, and effective interventions to support this carer population. I am not sure 'carers' would necessarily define themselves as such. It is one role among a number. It seems a little bit of a large aim, and slightly unspecified. The focus might be a little overly medical. What does it mean to care? A reflection on the wider dynamics around caring might help. It is probably not a unidimensional issue for most carers, there might be gender role issues etc.

We agree that the term 'carer' is problematic because many people who provide unpaid care do not identify themselves as such. Carers UK, the advocacy and support organisation for those providing unpaid care, highlights this point in relation to being able to identify this population in order to provide support (see Missing Out: the Identification Challenge, Carers UK, 2016). However, 'carers' is the term used within UK policy and practice to refer to those providing unpaid care, and we mirror this in the paper.

We have added the following to the first paragraph in the introduction, to acknowledge this tension:

In this paper, we use the term 'carer' to refer to people providing unpaid care, although we recognise that not every carer identifies themselves as such.

The overarching aim of our work was broad as we intended to map evidence about what is known and key gaps. However, as stated in the introduction and in the wording of our key questions, we have specified that the focus of our work was on the consequences of caring and interventions. We have provided details to ensure that our 'key questions' are specific; in particular, looking for evidence on consequences for different aspects of health, social and financial wellbeing, and how consequences vary by age, sex, socioeconomic status and geographical location. These details are also mirrored in the subsequent findings and discussion.

We agree that wider dynamics in relation to caring is important and we have avoided a focus on medical consequences. We specifically looked for evidence about the impact on health, but also on **social and financial wellbeing**. The available evidence was largely related to health, but this is more a reflection of current trends in available evidence. Indeed, in our discussion we advocate that supporting carers should include supporting their social and financial wellbeing. Furthermore, our location of

unpaid caring within an existing framework of person-level, socioeconomic, structural and cultural determinants (using the Rainbow model) also avoids the medicalisation of caring.

We do agree that caring is not unidimensional, and this is partly why we looked for evidence about how the impact of caring might differ between groups. We have added the following to the discussion to further emphasise this point:

Furthermore, carers are not a homogenous population, and unpaid caring takes place in diverse circumstances. A greater focus on which groups of carers are most vulnerable to the impact of caring, and under which conditions, would further enhance our understanding of these issues.

Reviewer 2: no comments given	
Reviewer 3	
This is an important topic, particularly in light of the pandemic, when more unpaid carers have to provide unpaid care for older adults, with a lack of social care services available. I only have a few comments, however, the last comment (all in chronological order) is major and needs specific attention:	Thank you for your constructive review of our paper. We have made revisions, and summarise these and our responses below. Edits are in red text.
- The title should reflect that this is a review of SRs.	We have amended the title also in line with the Editor's requirements: What are the consequences of caring for older people and what interventions are effective for supporting unpaid carers? A rapid review of systematic reviews

- Old age is classed as 65 and above (although that is incredibly outdated as we know...). however, please indicate why you have referred to age 85+ in the Introduction when referring to older adults.	We agree that the threshold of age 65 is an arbitrary and outdated marker of what is considered 'old' age. However, this threshold was used as a pragmatic consideration of how 'old' age is often operationalised. However, we appreciate the inconsistency between our criteria and the opening paragraph, and we have revised the content to reflect this: Populations are ageing worldwide. Life expectancy is increasing in many high-income countries, whilst the proportion of those aged 65 and over is projected to increase 16% by 2050.² However, longer lives will not necessarily be spent in good health, and population ageing will likely result in an increase in care needs.
- There is a grammar mistake at the end of the intro "We *under* undertook....".	Thank you. Corrected.

- It seems strange that carers of people living with dementia were excluded from the study (specified under Public involvement). Could the authors please elaborate on this? Who made up the PPI group? Did it include carers, including those with dementia?

During our review we identified three recently published umbrella reviews of systematic reviews on caring for people with dementia. The existence of these reviews prompted discussion with the project steering group about the focus of our work. To avoid duplicating these recent umbrella reviews, we focused our work on evidence about carers for older populations *not* specific to those with dementia. We have supplemented our explanation of this in the methods with the following sentence:

In consultation with a study steering group, reviews that focused solely on carers of people with dementia were excluded, as evidence about dementia caregivers has been summarized in three recently published reviews.²⁰⁻²² This exclusion criterion avoided duplication of this recent work and ensured we addressed the research questions for carers of older populations more broadly.

The project steering group included representatives from Carers UK, adult social care (England Local Authorities) including carer leads, and from Public Health England.

- The authors very briefly mention Dahlgren and WhiteHEAD (spelling mistake again)'s model of health inequalities, yet fail to elaborate how this is relevant to the topic and the findings. The authors should elaborate on this further and thus strengthen this point.

We have added the following (edit in red text) to elaborate our point:

One mechanism to achieve this is by conceptualising unpaid caring as a social determinant of health. Social determinants are factors relating to life conditions and circumstances, such as age, income, and working conditions, that impact on health. To locate unpaid caring as a social determinant recognises that this responsibility is likely to place carers at greater risk of poor health. Conceptualising unpaid caring as a social determinant of health is possible with an adaption of Dahlgren and Whitehead's (1991) model (figure 2).

The spelling mistake is also corrected, thank you

- If a more robust evidence base is needed, as outlined by the authors, how come a research priority is to conduct yet another SR? This would suggest more primary research is needed, to strengthen that evidence base. This seems particularly unnecessary as the authors state repeatedly that the included reviews were mostly conducted in the past few years, and thus conducting yet another SR would be a duplication of effort.

One of the three future research priorities suggested is a systematic review (the remaining two being primary research). We made this suggestion in recognition that current reviews fail to capture a broad range of outcomes (typically focusing on mental health) and to exploit existing grey literature, surveys and qualitative research that current reviews do not include. We apologise that this was not made clear, and we have amended this section to clarify:

1. A high quality comprehensive systematic review about the impact of caring for older people on outcomes overlooked in current systematic reviews: physical health, and social and financial wellbeing. A systematic review of these outcomes should draw upon evidence from non-peer reviewed sources (for example, national statistical reports, third sector reports) and qualitative research to capture the full breadth of available evidence.